

# Immune-related hub genes in intrauterine adhesions: a bioinformatics approach

Fengqing Lv[1], Sang Luo[2], Fengjuan Xu[1], Yue Du[2], Yiyun Bai[1], Jingyi Zhang[1], Xiaojie Zou[1] and Dan Liu[2,3,4]

[1] College of the First Clinical Medicine, Ningxia Medical University, Yinchuan, Ningxia, China
[2] Institute of Medical Sciences, General Hospital of Ningxia Medical University, Yinchuan, Ningxia, China
[3] Department of Gynecology, General Hospital of Ningxia Medical University, Yinchuan, Ningxia, China
[4] Key Laboratory of Ministry of Education for Fertility Preservation and Maintenance, General Hospital of Ningxia Medical University, Yinchuan, Ningxia, China

Corresponding author
Dan Liu, nxld@nyfy.com.cn

## ABSTRACT

**Background**. Intrauterine adhesions (IUA) are a leading cause of acquired female infertility that predominantly arises following surgical intrauterine interventions. Clinical strategies are available for managing IUA, however, the molecular pathogenesis of IUA, particularly the role of immune dysregulation in endometrial repair processes, has not been fully characterized, necessitating comprehensive mechanistic studies.

**Methods**. We used computational biology methods to determine the molecular pathogenesis of IUA, the results of which were experimentally validated. (i) We systematically reanalyzed GSE224093, a publicly available endometrial transcriptomic dataset, using GEO2R. Differential gene expression was analyzed with stringent statistical thresholds; the results were immunologically contextualized *via* intersection with the ImmPort database. (ii) Multilayer functional annotation was conducted using Gene Ontology (GO) enrichment and Kyoto Encyclopedia of Genes and Genomes (KEGG) pathway topology analyses. (iii) Weighted gene co-expression network analysis and scale-free topology optimization were used to identify conserved coexpression modules. (iv) Seven hub genes underwent quantitative real-time polymerase chain reaction (qPCR) validation in human and murine models, with four verified using immunohistochemistry (IHC).

**Results**. Our integrated multiomics analysis identified seven hub genes associated with IUA pathogenesis through GO, KEGG, and GSEA enrichment analyses. The expression levels *TUBB3*, *WNT5A*, *GDF7*, *IGF1*, and *BIRC5* were downregulated, and those of *PTGDS* and *CCL14* were upregulated. The qPCR results confirmed these expression patterns in human and murine endometria ($p < 0.05$), with *TUBB3* and *PTGDS* species-specifically diverging from the computational predictions. The IHC results provided the corresponding protein expression changes for IGF1, WNT5A, BIRC5, and GDF7 in IUA ($p < 0.01$). CCL14 could not be amplified in murine models due to technical constraints.

**Discussion**. We identified seven immune-related gene signatures through integrative bioinformatics. We experimentally validated three genes (*TUBB3*, *PTGDS*, and *CCL14*) demonstrating species-specific expression patterns. We proposed four mechanistically plausible biomarkers (WNT5A, IGF1, BIRC5, and GDF7) for developing IUA diagnostic tools. The conserved dysregulation of WNT5A- and TGF-β-associated genes (*GDF7* and *IGF1*) suggest therapeutic targets for preventing adhesion recurrence. This study advances our understanding of IUA pathogenesis. Single-cell transcriptomics should

be examined in future studies to determine the cellular-subtype-specific contributions to IUA.

## INTRODUCTION

Intrauterine adhesions (IUA) are pathological conditions, characterized by endometrial fibrosis; IUA is commonly known as Asherman's syndrome. IUA typically arises from trauma to the basal layer of the endometrium caused by infection, curettage, or other uterine manipulations, which disrupts the normal regenerative capacity of endometrial stromal cells and triggers aberrant tissue repair (*Khan, 2023*; *Torres-De La Roche, 2019*). The resulting fibrosis in IUA leads to partial or complete obliteration of the uterine cavity, which is often accompanied by endometrial thinning or atrophy. IUA clinically manifests as hypomenorrhea, infertility, and recurrent pregnancy loss, substantially compromising reproductive health (*Di Guardo et al., 2020*). The current therapeutic strategies for IUA primarily focus on surgical adhesiolysis to restore uterine anatomy, involving adjuvant measures such as mechanical barriers (*e.g.*, intrauterine devices or balloons) and hormonal therapy to prevent readhesion (*Dreisler & Kjer, 2019*). However, the postoperative recurrence rates of IUA remain high (*Salazar, Isaacson & Morris, 2017*), primarily due to persistent inflammation, impaired angiogenesis, and dysregulated immune responses that promote fibrosis and hinder functional endometrial regeneration. The failure of these therapies to address the underlying immune microenvironment dysregulation—including aberrant macrophage polarization, excessive cytokine production (*e.g.*, TGF-β), and disrupted cross-talk between immune and stromal cells—limits their long-term efficacy. Targeting these immunological imbalances could mitigate fibrosis recurrence and increase endometrial repair (*Ong et al., 2021*; *Queckbörner et al., 2020*; *Liu et al., 2019*). Thus, the pathophysiological mechanisms underlying the immune dysregulation in IUA should be determined to guide the development of novel strategies to reduce recurrence rates and restore reproductive function.

IUA results from the dysregulation of the local immune microenvironment following uterine trauma, impairing endometrial repair processes (*Lee et al., 2021*). This dysregulation involves aberrant immune activation after injury, dysregulated interactions among inflammatory mediators, and the immune cells participating in key pathological processes, such as epithelial–mesenchymal transition (EMT), abnormal cellular proliferation, and fibrosis. We identified immune-related differentially expressed genes (DEGs) associated with IUA from the Gene Expression Omnibus (GEO) database using bioinformatic analysis. Weighted gene co-expression network analysis (WGCNA) was used to identify the hub genes closely that are linked to IUA progression. The results of the subsequent functional analysis revealed the critical molecular pathways and candidate genes implicated in IUA pathogenesis. The results of our integrative approach offers insights

into the mechanistic processes underlying IUA and may be used to guide the development of targeted therapeutic strategies (*Kulasingam & Diamandis, 2008*).

## SYSTEMS AND METHODS

### Study design

We implemented four-phase analytical framework to study the pathogenesis of IUA. (1) The results of primary screening identified IUA-associated DEGs using the GSE224093 database using the following criteria: $|\log_2 FC| \geq 1$ and Benjamini–Hochberg (BH) false discovery rate (FDR)-adjusted $p < 0.05$. We independently curated the immune-related genes (IRGs) from ImmPort. (2) Functional convergence analysis was conducted using Venn intersection to identify overlapping gene. This analysis was followed by characterizing these genes through Gene Ontology (GO) enrichment analysis, Kyoto Encyclopedia of Genes and Genomes (KEGG) pathway analysis, Sankey network visualization, and gene set enrichment analysis (GSEA) using the KEGG gene sets. (3) We used weighted gene co-expression network analysis WGCNA to identify the conserved IUA modules, the results of which were confirmed through eigengene heatmaps and DEG-IRG correlation bar plots. (4) The seven identified hub genes were experimentally verified through qPCR in human and murine models, validating four genes (*IGF1, WNT5A, BIRC5,* and *GDF7*) that showed complete transcriptional agreement ($p < 0.05$, Pearson's $r > 0.85$). Protein-level results were obtained with immunohistochemistry (IHC) (Fig. 1). This integrated approach combined bioinformatic methods with experimental validation, providing comprehensive molecular insights into the development of IUA.

### Data sources and DEG screening

GSE224093, an IUA microarray dataset, was retrieved from the GEO database. The dataset comprises endometrial samples from seven patients with severe IUA and seven normal controls, profiling 33,944 genes. All data are publicly accessible.

We used the GEO2R optimized Limma pipeline to ensure robust and unbiased differential gene expression analysis of the GSE224093 microarray dataset as well as address the inherent data challenges, including the nonunique gene identifiers and Fragments Per Kilobase of transcript per Million mapped reads (FPKM)-normalized values incompatible with count-based methods. This platform-specific approach avoided the need for arbitrary gene filtering with DESeq2, which requires unique gene symbols and integer counts, as well as problematic data transformations for Limma-voom applications. We thus preserved potentially significant biological signals. The analysis involved (1) automated probe-to-gene annotation using platform-specific GEO Platform Library (GPL) files; (2) quantile normalization with low-expression filtering, retaining features with log2 intensity >6 in ≥50% of samples; and (3) linear modeling with empirical Bayes moderation (eBayes) to assess group-wise differences and adjust for heteroscedasticity. Differential gene expression was determined using two stringent thresholds, $|\log2FC| \geq 1$ with Benjamini–Hochberg (BH) FDR-adjusted $p < 0.05$. The quality of the results was controlled using diagnostic volcano plots. The top 100 significant DEGs were visualized *via* hierarchical clustering in Multiexperiment Viewer (MeV).
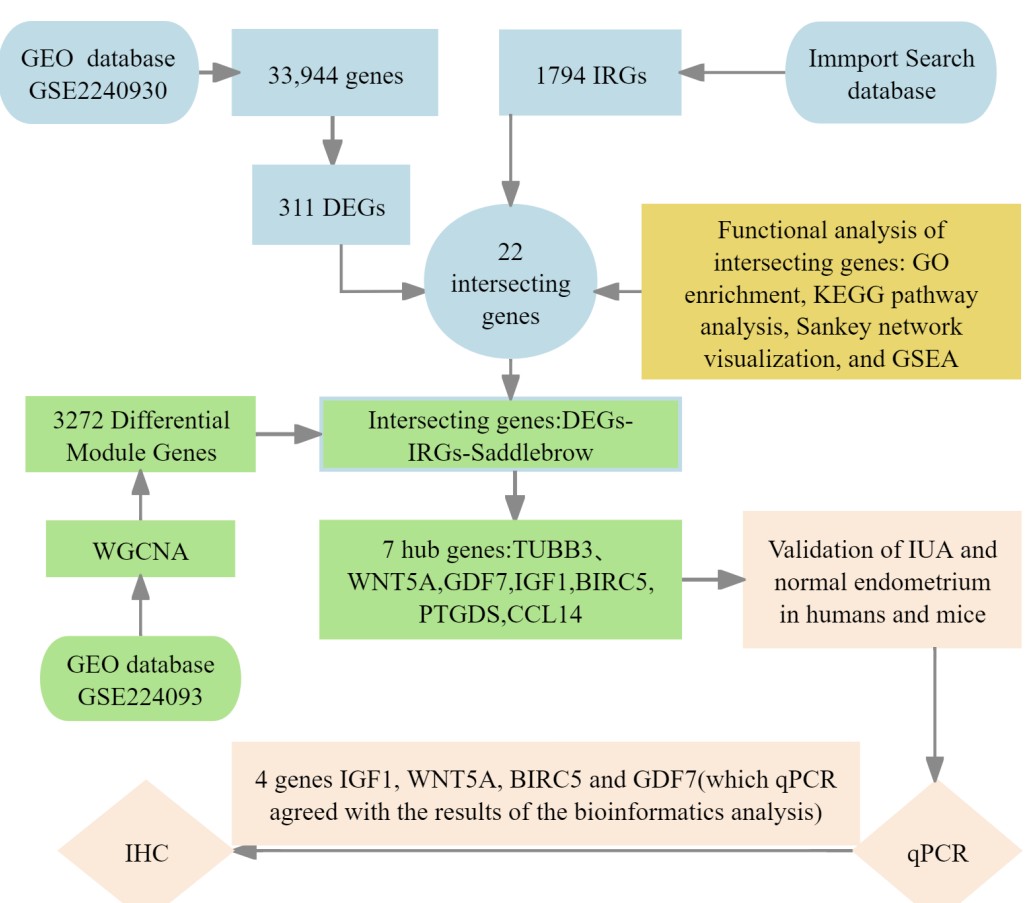

**Figure 1 Flowchart of bioinformatics analysis and clinical validation for intrauterine adhesions.**

## Identifying IRGs and intersecting genes

We obtained a set of 1,794 IRGs from the ImmPort database to identify the IRGs associated with IUA. We used an online bioinformatics platform (http://www.bioinformatics.com.cn/) to intersect these IRGs with the DEGs identified from IUA samples. The results identified 22 overlapping genes as immune-related candidates involved in the pathogenesis of IUA.

## Functional enrichment analysis of intersecting genes

We performed GO and KEGG pathway enrichment analyses to characterize the biological functions and pathways associated with the DEGs. The GO (http://www.geneontology.org/) and KEGG pathway (http://www.genome.jp/kegg/pathway.html) analyses were conducted using the Database for Annotation, Visualization, and Integrated Discovery (DAVID; https://davidbioinformatics.nih.gov) (*Givant-Horwitz, Davidson & Reich, 2005*). Terms with a threshold of $p < 0.05$ were considered significantly enriched. Additionally, the Metascape database (https://metascape.org/) was used to annotate and functionally cluster the DEGs and intersecting IRGs involved in IUA.

## Screening and verifying hub genes

We applied WGCNA on the GEO datasets to additionally validate the results. We first preprocessed the data through calculating the median absolute deviation and removing outliers using the WGCNA goodSamplesGenes method. We then constructed a scale-free coexpression network by computing Pearson's correlation matrices. We established a weighted adjacency matrix ($A_{mn} = |C_{mn}|^{\beta}$, $\beta = 12$), which we transformed into a topological overlap matrix (TOM). We identified gene modules using TOM-based dissimilarity (1-TOM) with average linkage hierarchical clustering (minimum module size = 30, sensitivity = 3). We merged the gene modules with a dissimilarity of <0.25, identifying the significant saddlebrown module, containing 3,272 genes. The highly connected hub genes in this module were intersected with the DEGs and IRGs (DEGs–IRGs–saddlebrown), from which we selected the seven most significantly dysregulated genes ($|\log 2FC| \geq 1$, BH-FDR $p < 0.05$) whose expression patterns consistently matched our preliminary DEG analysis. We thus confirmed their identification as hub genes in IUA pathogenesis.

## Correlation analysis between gene expression and immune cell infiltration

We conducted computational immunogenomic analysis to characterize the immune microenvironment in IUA and investigate the relationship of this microenvironment with gene expression patterns. The normalized gene expression matrix from all training cohort samples was analyzed using CIBERSORTx (version 1.0) with the LM22 signature matrix and 1,000 permutations, disabling quantile normalization to accurately deconvolute the relative proportions of 22 functionally distinct immune cell subsets. We controlled data quality by excluding samples with CIBERSORTx $p$-values $\geq 0.05$. Correlation analysis was performed using Spearman's rank correlation coefficient, with multiple testing correction *via* the BH method (FDR < 0.05 considered significant). We used deconvolution algorithms (xCell and Microenvironment Cell Populations (MCP)-counter) for comparative analysis to validate the robustness of our findings. This multialgorithm approach allowed us to comprehensively assess immune cell infiltration patterns and their potential associations with gene expression profiles in the pathogenesis of IUA.

## qRT-PCR
### RNA extraction and quality control

All procedures were performed on ice to prevent RNA degradation. The total RNA was extracted from 40–50 mg tissue samples from the normal and IUA groups using TRIzol reagent following the protocol provided by the manufacturer. The RNA concentration and purity were assessed using NanoDrop spectrophotometry, with triplicate measurements averaged for accuracy. Only samples with A260/A280 ratios between 1.8 and 2.2 were considered for downstream analysis. Purified RNA aliquots were stored at −80 °C until use.

### Genomic DNA elimination

A total of one $\mu$g of total RNA from each sample was treated in a 10 $\mu$L reaction volume containing 2.0 $\mu$L of 5$\times$ gDNA Eraser Buffer, 1.0 $\mu$L of gDNA Eraser, and RNase-free dH2O. The sample was incubated at 42 °C for 2 min, which was followed by immediate cooling to 4 °C to ensure the genomic DNA was completely removed and RNA integrity was preserved.

### Complementary DNA (cDNA) synthesis

Reverse transcription was performed in 20 $\mu$L reactions comprising 1.0 $\mu$L of PrimeScript RT Enzyme Mix I, 5.1 $\mu$L of RT Primer Mix, 4.0 $\mu$L of PrimeScript Buffer 2, and 4.0 $\mu$L of RNase-free (dH$_2$O), using the following thermal program: 37 °C for 15 min (reverse transcription), 85 °C for 5 s (enzyme inactivation), and a 4 °C hold. The synthesized cDNA products were aliquoted and stored at $-40$ °C until downstream application.

### qPCR amplification

The key hub genes (*TUBB3, WNT5A, GDF7, IGF1, BIRC5, PTGDS,* and *CCL14*) were amplified using Primer3-designed primers (Table 1) in 20 $\mu$L reactions containing 10 $\mu$L of TB Green Premix Ex Taq II, 0.8 $\mu$L each of the forward and reverse primers (10 $\mu$M), two $\mu$L of cDNA template, and 6.4 $\mu$L of RNase-free dH2O. Amplification was conducted with a qTOWER3/G system (Analytik Jena) and the following protocol: initial denaturation at 95 °C for 30 s (one cycle), which followed by 40 cycles of amplification (95 °C for 5 s, 60 °C for 34 s), and melt curve analysis (95$\rightarrow$ 60 $\rightarrow$95 °C, 15 s per step). Each 96-well plate contained quadruplicate samples from the IUA and control groups for three target genes using Glyceraldehyde-3-Phosphate Dehydrogenase (GAPDH) as the reference, with a minimum of three technical replicates per biological sample.

### qPCR data analysis and quality control

The reaction specificity was confirmed *via* melt curve analysis. The data were normalized using the 2^-$\Delta\Delta$Ct method using GAPDH as the reference. Samples were excluded based on predefined criteria: multiple melt curve peaks, asynchronous peaks *versus* reference, Ct values <18 or >32, or intrasample Ct variation >0.5. The $\Delta$Ct was calculated as Ct(target)–Ct(reference). $\Delta\Delta$Ct was calculated as $\Delta$Ct(experimental)–$\Delta$Ct(control). The relative expression was calculated using the 2^-$\Delta\Delta$Ct method.

## IHC

The IUA and normal endometrial tissue samples were fixed in 10% neutral buffered formalin, embedded in paraffin, and sectioned into four $\mu$m thick slices. The samples were deparaffinized and rehydrated. Then, antigens were retrieved using citrate buffer (pH 6.0). Nonspecific binding sites were blocked with 5% bovine serum albumin solution.

The primary antibodies were incubated using a rabbit monoclonal antibody specific to the target protein. The optimal antibody concentration was determined through preliminary experiments on human and mouse endometrial tissues, including the IUA and control groups, in accordance with the manufacturer's protocol. This predetermined concentration was then applied to all study samples.

**Table 1 Gene accession numbers and primer sequences for qPCR analysis.**

| Target gene | Species | Target gene sequence accession number | Primer | primer sequence (5′to 3′) | Number of bases |
|---|---|---|---|---|---|
| IGF1 | Mice | NM010512 | F | GCTCTGCTTGCTCACCTTCACC | 22 |
| | | | R | CGGTCCACACACGAACTGAAGAG | 23 |
| | Human | NM000618 | F | TGTCCTCCTCGCATCTCTTCTACC | 24 |
| | | | R | CCTGTCTCCACACACGAACTGAAG | 24 |
| WNT5A | Mice | NM009524 | F | AAGGATTTCGTGGACGCTAGAG | 22 |
| | | | R | GCCTGCATTGTTGTGTAAGTTC | 22 |
| | Human | NM003392 | F | GACTTCCGCAAGGTGGGTGATG | 22 |
| | | | R | GTCTTGTGTGGTGGGCGAGTTG | 22 |
| BIRC5 | Mice | NM009689 | F | TCATCCACTGCCCTACCGAGAAC | 23 |
| | | | R | CGGGTTCCCAGCCTTCCAATTC | 22 |
| | Human | NM001168 | F | TCTCAAGGACCACCGCATCTCTAC | 24 |
| | | | R | CCAAGTCTGGCTCGTTCTCAGTG | 23 |
| GDF7 | Mice | NM008117 | F | GCCACACCACTTCATGATGT | 20 |
| | | | R | TGAAGCCGGTGATTGTGTCC | 20 |
| | Human | NM182828 | F | GCCGCACCACTTCATGATGTC | 21 |
| | | | R | CTGTGAAGCCGGTGATCGTGT | 21 |
| TUBB3 | Mice | NM023279 | F | GAAGCCCTCTACGACATCTG | 20 |
| | | | R | TTGAGCTGACCAGGGAATCG | 20 |
| | Human | NM006086 | F | TGCGGAAGGAGTGTGAAAAC | 20 |
| | | | R | GATACTCCTCACGCACCTTG | 20 |
| PDGDS | Mice | NM008963 | F | TTTGGTCCTCCTGGGTCTCTTGG | 23 |
| | | | R | CTTGTTGAAAGTTGGGCTGCACTG | 24 |
| | Human | NM000954 | F | GGAGAAGAAGGCGGCGTTGTC | 21 |
| | | | R | TGAGGAAGGTGGAGGTCAGGTTG | 23 |
| CCL14 | Mice | − | F | ACCTACATTACCCACGAGCT* | 21 |
| | | | R | GAAGACAATCCCAGGCTTG* | 20 |
| | Human | NM001001445 | F | CCTTACCACCCCTCAGAGTG | 20 |
| | | | R | TGGAGCACTGGCTGTTGGTC | 20 |

**Notes.**

⁻Dashes (−) indicate data not explicitly provided in the original text.
*Asterisks (*) indicate CCL14 was included in primer design but yielded no quantifiable qPCR results.

We incubated the samples with a horseradish peroxidase (HRP)-conjugated antirabbit IgG secondary antibody, which was followed by chromogenic development with 3,3′-diaminobenzidine. The sections were counterstained, dehydrated, and mounted with neutral resin for microscopy evaluation.

All immunohistochemical slides were independently reviewed by a board-certified pathologist. The staining intensity was quantitatively analyzed using ImageJ software. The results are expressed as the percentage positive area (% Area) for statistical comparison.

## Sample sources and primary reagents

We confirmed the hub gene differential expression in clinical and mice cohorts. The sample sizes across all validation platforms (qPCR: humans, 4–8/group; mice, 4–6/group. IHC: humans, 5–8 IUA *vs.* five controls; mice, five IUA *vs.* five controls) were determined considering the statistical requirements and practical constraints. We incorporated (1) prospective power calculations (G*Power 3.1), anticipating large effect sizes (Cohen's $d \geq 1.2$ from preliminary bioinformatics analyses), which confirmed 72–85% power in detecting significant differences ($d \geq 1.0$, $\alpha = 0.05$ two-tailed) after BH correction; (2) quality control thresholds (RNA Integrity Number (RIN) $\geq 7$ for qPCR) that inherently limited specimen inclusion; (3) alignment with established methodological standards for nonparametric analyses (*Wittwer et al., 2009*); (4) ethical and logistical limitations in clinical specimen acquisition. These sample sizes enabled the robust detection of large-effect biomarkers as evidenced by the consistent confirmation of the hub genes; however, we acknowledge the low sensitivity in identifying moderate effects ($d = 0.5-0.8$), a limitation mitigated through our multiplatform validation approach. As such, larger cohorts should be included in future studies to comprehensively characterize subtle regulatory differences.

### Human samples

This study was approved by the Ethics Review Committee of the General Hospital of Ningxia Medical University (approval No. KYLL-2024-1527). All participants provided written informed consent prior to sample collection.

### Study population

We enrolled 24 patients with moderate-to-severe IUA and 13 control subjects with a normal endometrium. IUA severity was classified according to the American Fertility Society scoring system (*AAGL Advancing Minimally Invasive Gynecology Worldwide, 2010*). Participants were recruited from the Department of Gynecology at our institution between January 1 and September 1, 2024, aged 24–42 years.

### Endometrial sample collection

Endometrial specimens were concurrently obtained with postoperative pathological specimens during diagnostic hysteroscopy procedures, excluding patients aged $\geq 43$ years or with histopathological evidence of endometrial malignancy to ensure sample quality and homogeneity.

### Sample processing and allocating

All samples collected between January 1 and September 1, 2024, were processed following standard analytical protocols. qPCR samples (12 IUA and 8 control samples) were immediately snap-frozen and stored at $-80$ °C to preserve RNA integrity. The IHC samples (12 IUA and five controls) underwent 24-hour fixation in 10% neutral buffered formalin followed by standard paraffin embedding for histological analysis.

### Quality control

Four IUA specimens were excluded during qPCR processing based on inadequate nucleic acid quality metrics (purity and concentration thresholds). All IUA samples were stained

with hematoxylin–eosin (HE) staining for verification of quality prior to IHC. Four samples failing to meet the moderate/severe IUA criteria were excluded.

### Final sample

This study included two experimental cohorts: (1) a qPCR analysis cohort comprising eight IUA cases and eight controls and (2) an IHC validation cohort consisting of eight IUA cases and five controls. All samples met our strict quality control criteria prior to analysis.

### Animal experiment

*Ethical approval and animal housing.* All experimental procedures were conducted in accordance with protocols approved by the Institutional Animal Care and Use Committee (IACUC) of the General Hospital of Ningxia Medical University (approval No. IACUC-NYLAC-2023-136). Specific-pathogen-free-grade female C57BL/6J mice (6–8 weeks old, 17–20 g) were obtained from the Laboratory Animal Center of Ningxia Medical University. The mice were maintained under controlled conditions: 20–26 °C, 50–60% humidity, 12-hour light/dark cycle, and housed in independent ventilation cages ($\leq 5$ mice per cage) with same-age female cohorts. The cages were changed weekly.

*Experimental group allocation.* The mice were randomly assigned to either (1) a control (CON) group ($n = 5$), receiving no surgical intervention, or (2) an IUA model group ($n = 8$), in which endometrial lesions were surgically induced.

*Surgical modeling (IUA Group).* The mice were anesthetized *via* the intraperitoneal injection of 1.25% tribromoethanol (two mL/100 g body weight), which was followed by aseptic surgical preparation. Midline laparotomy was performed to expose the uterine horns, and a 0.1 cm transverse incision was cut 0.5 cm proximal to the uterine bifurcation. Endometrial mechanical injury was induced using a sterile curette until visual confirmation of tissue congestion and edema. The uterus was carefully repositioned following the procedure, and the abdominal cavity irrigated with sterile saline. The surgical site closed in layers with absorbable sutures.

*Euthanasia and tissue collection.* The mice were euthanized seven days after modeling *via* an intraperitoneal injection of 1.25% tribromoethanol (six mL/100 g body weight). Death confirmed by the absence of vital signs (cessation of chest movement, eyelid blanching, and corneal reflex) and verified through another assessment 30 min after the procedure. Uterine tissues were bilaterally harvested, with one horn snap-frozen in liquid nitrogen until the subsequent qPCR analysis and the contralateral horn fixed in 10% neutral buffered formalin for IHC studies.

*Model validation and quality control.* We established success criteria involving evaluating macroscopic uterine morphology and histopathology by measuring endometrial thickness and quantifying glandular parameters *via* HE staining, referencing established standards (*Zhang et al., 2017*). Two IUA specimens were excluded due to not meeting the nucleic acid quality and/or concentration thresholds. The three IUA samples that failed to meet the

moderate/severe adhesion criteria during HE screening were excluded from the subsequent analyses to ensure model fidelity.

We used two independent sample sets for analysis: a qPCR cohort consisting of six IUA and five CON specimens as well as an IHC cohort comprising five IUA and five CON samples for protein-level validation.

### Primary reagents

*qPCR and Reverse Transcription.* cDNA was synthesized using a PrimeScript™ RT reagent kit (TaKaRa, Cat# RR047A). This was followed by qPCR amplification with TB Green™ Premix Ex Taq™ II (TaKaRa, Cat# RR820A) according to the manufacturer's protocols.

*IHC reagents.* The primary antibodies included *IGF1* (Proteintech, #28530-1-AP; human 1:300, mouse 1:800), *WNT5A* (Proteintech, #55184-1-AP; human 1:800, mouse 1:1200), *BIRC5* (ABclonal, #A1551; human 1:800, mouse 1:50), and *GDF7* (Abcam, #ab189928, Clone EPR16000; human/mouse 1:1500). Immunodetection was performed using HRP-conjugated antirabbit IgG secondary antibody (ZSGB-BIO, #PV-8000-1).

### Statistical methods

All statistical analyses were performed using GraphPad Prism 10 (GraphPad Software, USA). The data distribution was assessed through Shapiro–Wilk normality testing. Continuous variables were analyzed using either Student's $t$-test for normally distributed data or the Mann–Whitney U test for non-normally distributed data, with a two-tailed $p$-value $< 0.05$ considered statistically significant. The limited sample sizes (all $n < 10$) and non-normal distribution (according to the Shapiro–Wilk test results) necessitated the use of the Mann–Whitney U test in this study.

## RESULTS

### DEGs in IUA

The GSE224093 dataset was downloaded from the GEO website, comprising 33,944 genes. $|\log_2 FC| \geq 1$ and $p < 0.05$ were used as the filters to identify the DEGs. A total of 311 DEGs were obtained: 190 down- and 121 upregulated genes, respectively, and 33,633 genes that did not significantly differ in expression between the groups (Fig. 2A). The expression levels of the top 100 significant DEGs were analyzed (Fig. 2B). We then downloaded 1794 IRGs from Immport (http://www.immport.org/home). We used the intersection with DEGs in IUA, identifying 22 intersecting genes (Fig. 2C). We identified six downregulated genes through analyzing the expressions of the intersecting genes in the GSE224093 dataset (Figs. 2D, 2E): *BIRC5*, *LGR5*, *TUBB3*, *IGF1*, *GDF7*, and *WNT5A*. We also identified 16 upregulated genes: *ANGPTL1*, *ACKR1*, *EDNRB*, *NPR3*, *PTGDS*, *CCL14*, *PTGFR*, *APOD*, *CCL21*, *LYZ*, *CXCL10*, *PTGER3*, *PI15*, *FGF7*, *C3*, and *DES*.

### DEG functions and pathway enrichment

The genes overlapping among the IUA-associated DEGs and IRGs were functionally annotated and clustered using the Metascape database (https://metascape.org/). GO, KEGG pathway enrichment, and Sankey diagram analyses were performed *via* an

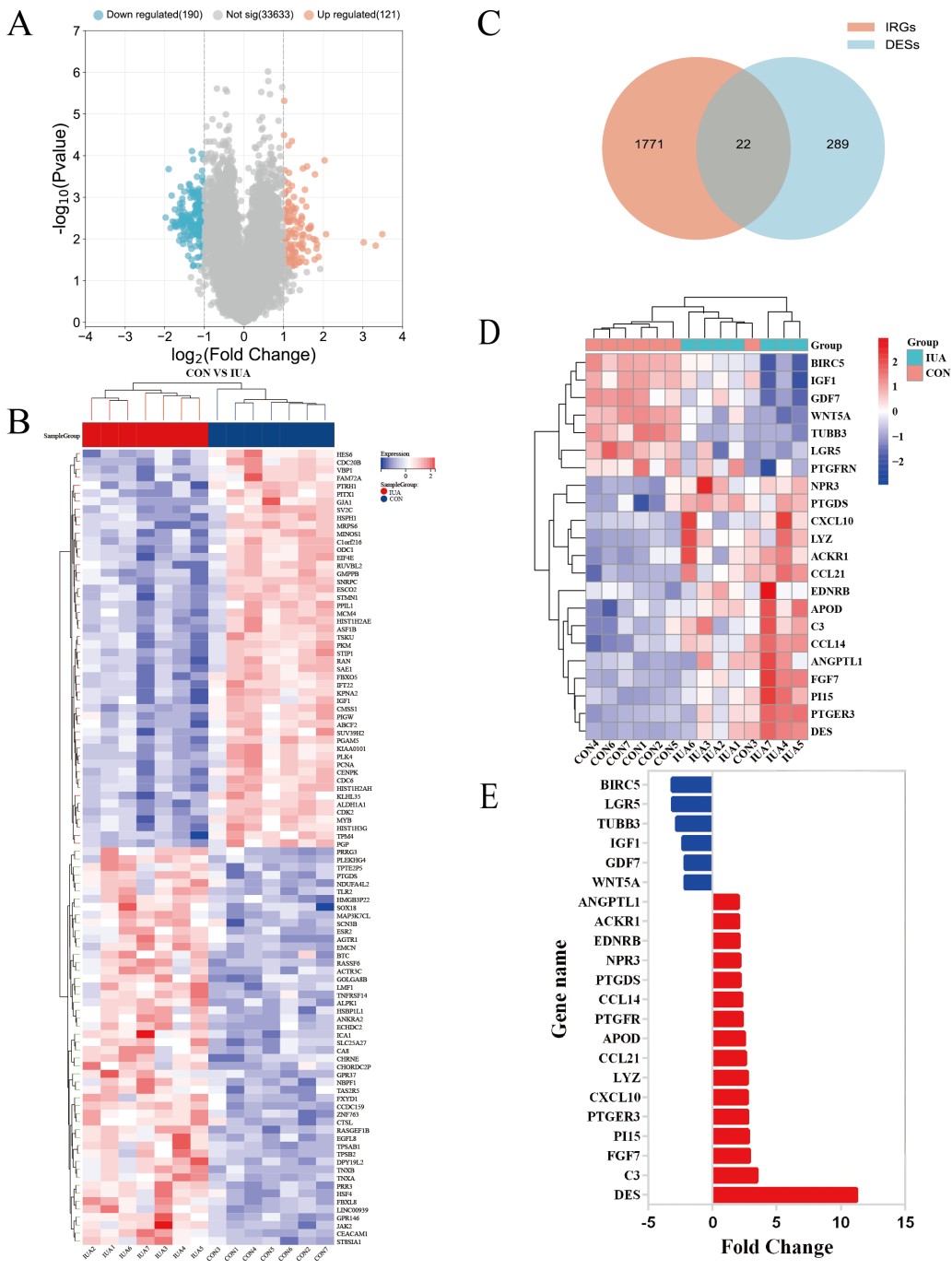

**Figure 2 Identification of differentially expressed genes and their overlap with immune-related genes in the GSE224093 dataset.** (A) Volcano plot of DEGs, with significantly up-regulated genes (red), down-regulated genes (blue), and non-significant genes (gray). (B) Hierarchical clustering heatmap of the top 100 DEGs, where red indicates higher expression and blue indicates lower expression. (C) Venn diagram illustrating the intersection between DEGs and immune-related genes in intrauterine adhesions. (D) Heatmap of intersected DEGs (red, up-regulated; blue, down-regulated). (E) Bar plot displaying log2 fold changes of key immune-related DEGs.

online bioinformatics platform (http://www.bioinformatics.com.cn/). The results of GO enrichment analysis revealed that the overlapping genes in the biological process category were predominantly associated with inflammatory response, chemotaxis, and protein phosphorylation. These results suggest a strong link between dysregulated immune processes and IUA pathogenesis (Fig. 3A). The results of KEGG pathway analysis demonstrated substantial enrichment of the DEGs in the Wnt signaling pathway, indicating a role in IUA progression (Fig. 3B). The Sankey diagram illustrated the gene–pathway relationships, showing that each gene mapped to 1–10 pathways, and each pathway contained one-six associated genes (Fig. 3C). GSEA was then conducted on the $|log_2FC|$-ranked gene list, using KEGG gene sets to identify broader pathway-level alterations. The GSEA results highlighted considerable enrichment in cell cycle regulation, oocyte meiosis, and spliceosome activity, with consistent activation and inhibition trends across these pathways (Fig. 3D).

## Gene validation

We performed WGCNA on the GEO database again to further screen the target genes. We analyzed the GEO database using a soft sign-independent threshold (Fig. 4A), an average continuity soft threshold (Fig. 4B), sample clustering (Fig. 4C), gene coexpression (Fig. 4D), a module feature vector (Fig. 4E), module phenotypic correlation (Fig. 4F), and a differentially expressed module (Fig. 4G). We obtained 3,272 differential module genes. We obtained the genes that intersected among the WGCNA differential module and DEGs, and IRGs (DEGs–IRGs–saddlebrow) (Fig. 4H). Seven significant genes were obtained, of which *TUBB3*, *WNT5A*, *GDF7*, *IGF1*, and *BIRC5* were downregulated and *PTGDS* and *CCL14* were upregulated (Figs. 4I, 4J).

## Clinical relevance of key genes

We identified seven key hub genes using bioinformatics methods and explored the molecular mechanisms underlying IUA: *TUBB3*, *WNT5A*, *GDF7*, *IGF1*, *BIRC5*, *PTGDS*, and *CCL14*. We conducted functional enrichment and pathway analyses to determine the roles of these genes in IUA. These genes are significantly enriched in biological processes related to inflammatory responses, chemotaxis, and the regulation of cytokine production, among other immune-related processes. These genes are implicated in several key signaling pathways, such as cytokine–cytokine receptor interactions, the PI3K/Akt signaling pathway, and the TGF-β signaling pathway, all of which may play roles in the development of IUA.

We performed GSEA to validate the involvement of these genes in IUA. These hub genes were found to be enriched in biological processes, such as the cell cycle, oocyte meiosis, and RNA splicing, suggesting that these genes influence the pathological mechanisms of IUA through affecting cell proliferation, differentiation, and RNA processing. We constructed a network map of the hub genes and their related signaling pathways, illustrating their complex interactions and their crucial roles in the inflammatory response and fibrosis. Among these genes, *TUBB3* is involved in the maturation of anatomical structures and is closely linked to cell cycle regulation, indicating its importance in the remodeling of IUA. The aberrant expression of *TUBB3* may contribute to the overproliferation and

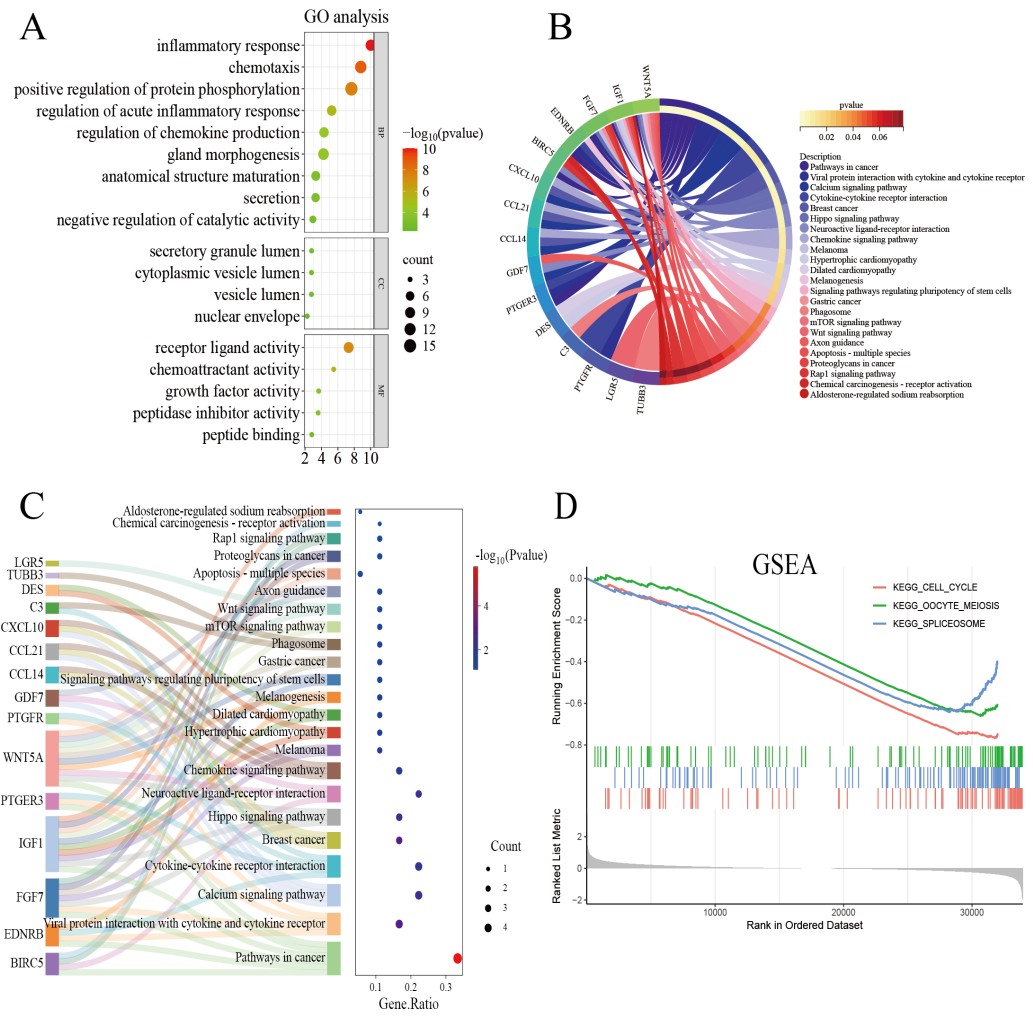

**Figure 3 Functional enrichment analysis of overlapping genes.** (A) Gene Ontology enrichment analysis of DEGs across biological processes, cellular components, and molecular functions. Dot size reflects gene count, and color scale represents statistical significance (–log10(*P*-value)). (B) KEGG pathway analysis of DEGs, highlighting pathway interactions. Color gradient indicates *P*-value significance (red, most significant; blue, least significant). (C) Sankey diagram illustrating associations between DEGs and enriched pathways. Left panel: pathway names; right panel: dot plot of gene ratios *versus* –log10(*P*-value). Connecting lines indicate gene-pathway relationships. (D) Gene Set Enrichment Analysis of DEGs, showing significant enrichment in cell cycle, oocyte meiosis, and spliceosome pathways. Enrichment score (*y*-axis) is plotted against ranked gene sets (*x*-axis), with vertical ticks marking gene positions in the expression profile.

fibrosis of uterine tissues (*Puri, Barry & Engle, 2023*). *WNT5A* was found to be enriched in the Wnt signaling pathway and inflammatory response-related GO terms, indicating roles in regulating inflammation and fibrosis (*Kikuchi et al., 2011*; *Pashirzad et al., 2017*). *GDF7* was found to be enriched in the TGF-β signaling pathway, suggesting involvement in tissue fibrosis (*Morikawa, Derynck & Miyazono, 2016*). IGF1 primarily participates in regulating growth factor activity and the PI3K/Akt signaling pathway, implying a role in IUA formation and progression through promoting cell proliferation and inhibiting

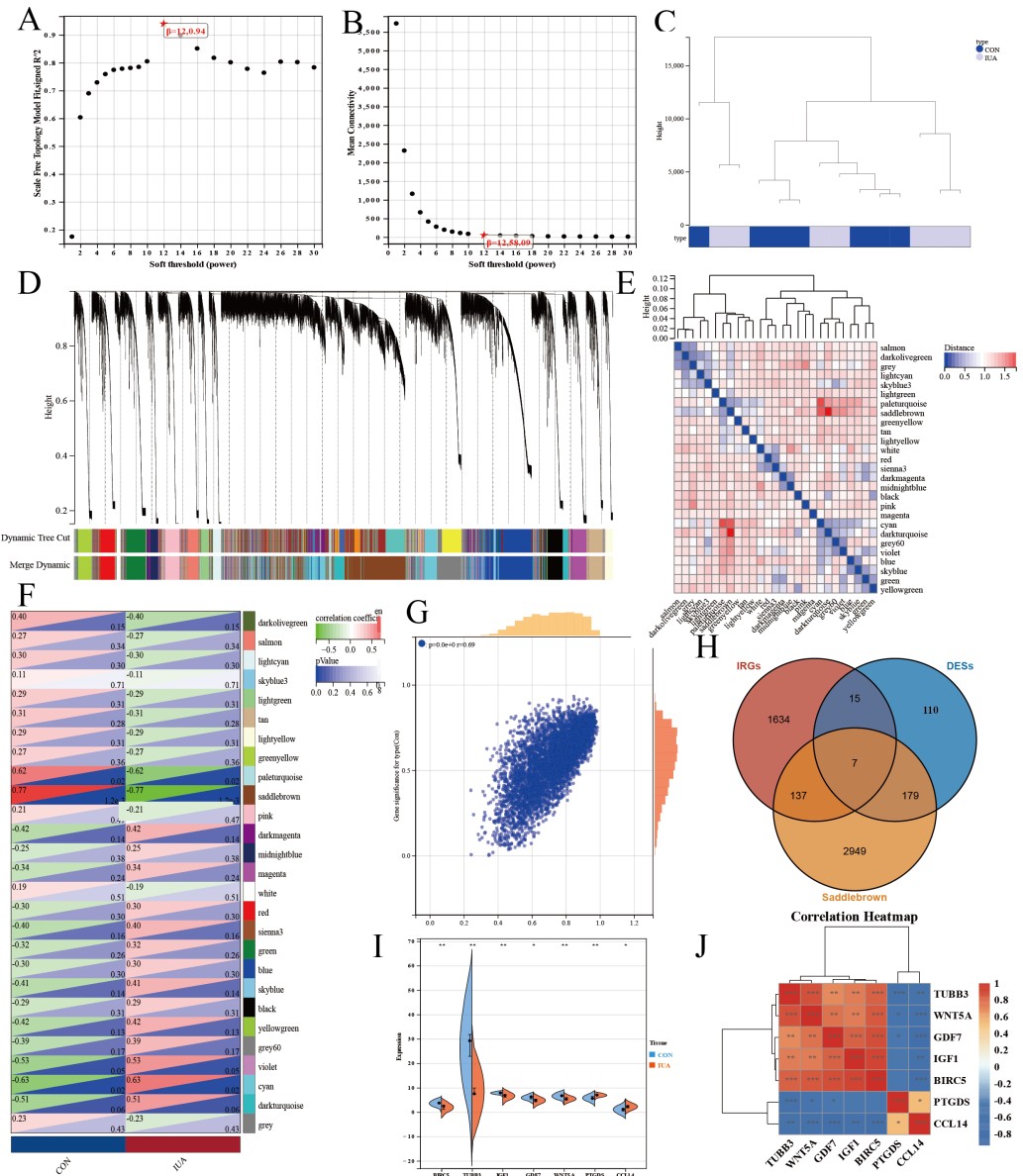

**Figure 4 WGCNA network analysis and hub gene identification in the GSE229043 dataset.** (A) Scale independence analysis for soft-threshold power selection. (B) Mean connectivity analysis showing the relationship between network connectivity and soft-thresholding power. (C) Sample clustering dendrogram with trait heatmap, demonstrating sample quality and potential outliers. (D) Gene co-expression module identification, where each color represents a distinct module. (E) Module eigengene dendrogram showing hierarchical clustering of identified modules. (F) Module-trait correlation heatmap, with color intensity representing Pearson correlation coefficients (red, positive; blue, negative). (G) Scatter plot of gene significance *versus* module membership for key phenotypic associations. (H) Venn diagram intersection of WGCNA module genes (saddlebrown), differentially expressed genes (DEGs), and immune-related genes (IRGs), identifying three consensus hub genes. (I) Differential expression validation of hub genes in intrauterine adhesion tissues compared to controls (\*\*\*$p < 0.001$, \*\*$p < 0.01$, \*$p < 0.05$). (J) Correlation heatmap of hub gene expression patterns, with color scale representing pairwise Pearson correlation coefficients (red, positive; blue, negative).

apoptosis (*Werner, 2023*). *BIRC5* was significantly enriched in anti-apoptosis-related GO terms, indicating a role in protecting IUA tissue through inhibiting apoptosis (*Sanhueza et al., 2015*; *Zhang et al., 2021*). *PTGDS* was found to be mainly associated with GO terms related to prostaglandin metabolic processes and inflammatory responses, suggesting involvement in the inflammation regulation (*Crookenden et al., 2023*; *Chandrasekaran, Weiskirchen & Weiskirchen, 2024*). Finally, *CCL14* plays a role in regulating chemokine activity regulation and recruiting immune cells, which was found to be enriched in GO terms related to the immune response and KEGG pathways, underscoring roles in the inflammatory response and immune regulation within the uterine cavity (*Allen et al., 2009*; *Zhu et al., 2019*).

The results of the GO, KEGG, and GSEA analyses together identified the potential roles of *TUBB3*, *WNT5A*, *GDF7*, *IGF1*, *BIRC5, PTGDS*, and *CCL14* in IUA. These hub genes may be integral to the pathological processes associated with IUA, influencing cell proliferation, inflammatory responses, fibrosis, and immune evasion.

### qPCR results

The results of qPCR validation revealed distinct expression profiles of the hub genes in the human and murine endometrial specimens (Fig. 5). IGF1, WNT5A, BIRC5 and GDF7 expression levels were significantly downregulated in the human IUA groups compared with the controls, whereas the expression levels of TUBB3, PTGDS, and CCL14 were upregulated. The TUBB3 expression level did not agree with the bioinformatic predictions. The expression levels of IGF1, WNT5A, BIRC5, GDF7, TUBB3, and PTGDS were similarly downregulated in the murine specimens. The measured PTGDS expression level in the murine specimens did not agree with the results of computational analysis. Technical limitations prevented the validation of CCL14 expression levels in the murine models due to primer unavailability and the potential gene absence in mouse endometrium. All statistical details are presented in Table 2.

### IHC results

The IHC results confirmed substantial differences in the IGF1, WNT5A, BIRC5, and GDF7 protein expression patterns between the normal and IUA groups in the human endometrial and murine tissues. The observed protein expression profiles completely agreed with the transcriptomic patterns identified through our biosignature analysis (Fig. 6). All statistical details are presented in Table 3.

## DISCUSSION

### Importance of immune microenvironment in formation of IUA

IUA is common gynecological disorder with complex pathomechanisms that are difficult to treat. We identified an important role of the immune microenvironment in the development of IUA. Immune imbalance impacts the endometrial fibrosis process (*Niu et al., 2023*), whereas *Chen et al. (2022)* further identified that CD4+ T-cell-mediated macrophage polarization plays a critical role in the formation of fibrosis. Our results

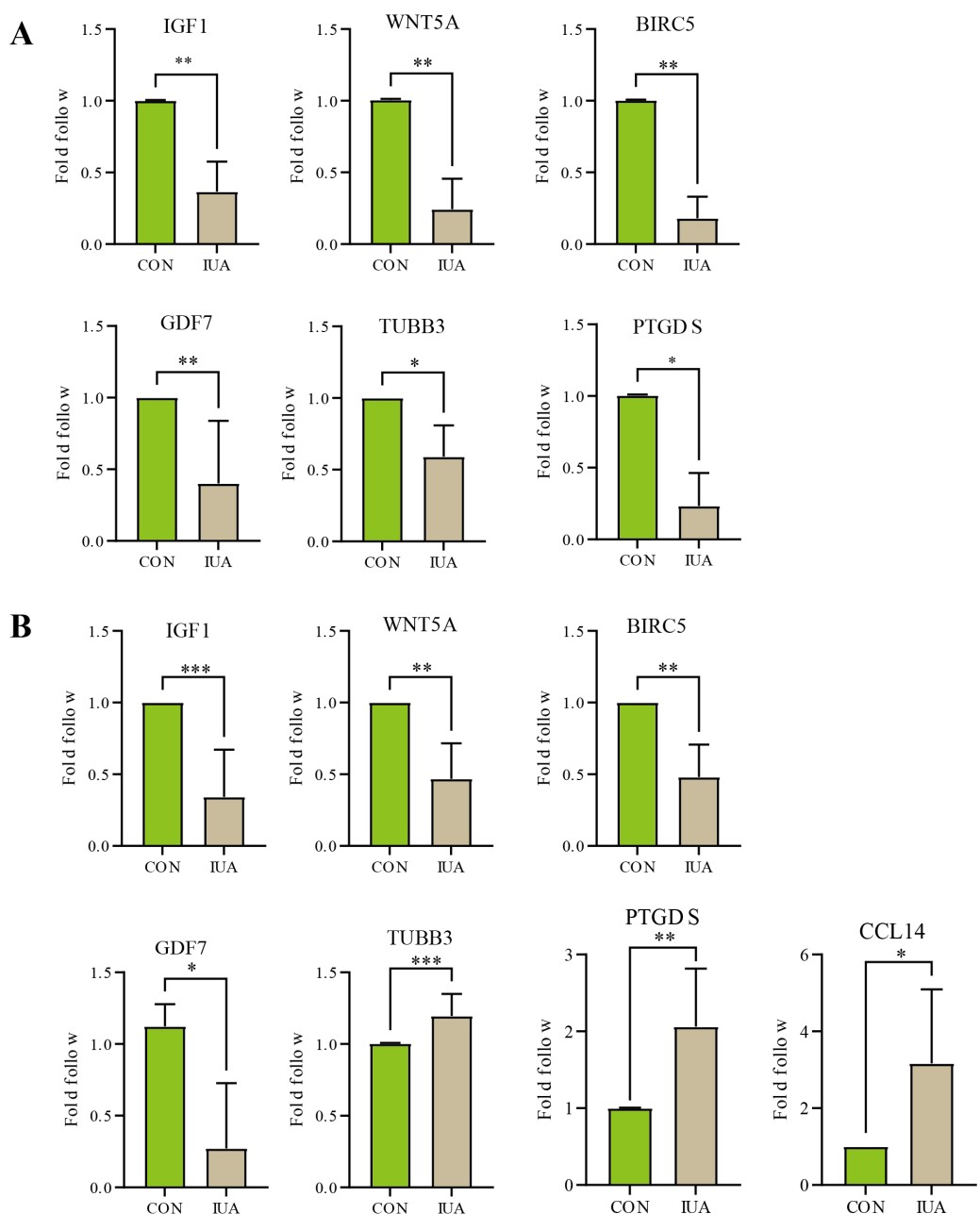

**Figure 5** **Validation of hub gene expression by qPCR.** (A) Relative mRNA expression levels of identified hub genes in endometrial tissues from a mouse model of intrauterine adhesions (IUA) compared with control mice. (B) Validation of hub gene expression in human endometrial tissues from IUA patients *versus* healthy controls. Statistical significance was determined by Mann–Whitney U test (***$p < 0.001$, **$p < 0.01$, *$p < 0.05$).

support these findings and demonstrate the changes in the expression of specific immune-related genes in IUA, which may be associated with immune cell activation, inflammatory factor release, and tissue repair processes (*Torres-De La Roche, 2019*).

**Table 2** Quantitative PCR analysis of target gene expression in human and murine endometrial tissues.

| Genes | Species | Sample size ($n_1$ *vs* $n_2$) | *P*-value | Cliff's delta [95% CI] |
|---|---|---|---|---|
| IGF1 | Human | 8 *vs* 8 | 0.0002 | −1[−0.95, −0.24] |
| | Mice | 6 *vs* 6 | 0.0022 | −1'[−0.89, −0.36] |
| WNT5A | Human | 6 *vs* 6 | 0.0022 | −1[−0.85, −0.23] |
| | Mice | 6 *vs* 6 | 0.0022 | −1[−1.00, −0.41] |
| BIRC5 | Human | 6 *vs* 6 | 0.0022 | −1[−0.77, −0.24] |
| | Mice | 6 *vs* 6 | 0.0022 | −1[−1.00, −0.58] |
| GDF7 | Human | 4 *vs* 4 | 0.0286 | −1[−1.34, −0.06] |
| | Mice | 5 *vs* 5 | 0.0079 | −1[−0.97, −0.09][*] |
| PTGDS | Human | 6 *vs* 6 | 0.0022 | −1[0.08, 1.91] |
| | Mice | 4 *vs* 4 | 0.0286 | −1[−1.01, −0.48] |
| TUBB3 | Human | 7 *vs* 7 | 0.0006 | −1[0.06, 0.25] |
| | Mice | 4 *vs* 4 | 0.0286 | −1[−0.73, −0.23][*] |
| CCL14 | Human | 4 *vs* 4 | 0.0286 | −1[0.06, 4.27][*] |
| | Mice | – | – | – |

**Notes.**
[n1] Normal control group.
[n2] Intrauterine adhesion (IUA) group.
[−]Dashes (–) indicate data not explicitly provided in the original text.
[*]Dashes (*) indicates 97% confidence interval (CI) (non-standard 95% CI).

## Role of hub genes in IUA

We identified key hub genes, such as *IGF1*, *WNT5A*, *BIRC5*, and *GDF7*, through bioinformatics analysis, which may play roles in regulating inflammation, immune activation, tissue repair, and fibrosis. *IGF1* is crucial for tissue repair, fibrosis, and immune responses. *IGF1* is involved in regulating tissue repair and fibrosis through the PI3K/Akt and TGF-β signaling pathways, promoting cell proliferation and matrix deposition in hepatic and cardiac fibrosis (*Wang et al., 2023*). *IGF1* downregulation is associated with aberrant fibrosis and impaired tissue repair, highlighting its potential as a therapeutic target (*Wang et al., 2018*). The roles of *WNT5A* in immune regulation, cell migration, and tissue remodeling are being increasingly supported. *WNT5A* is involved in pathological processes such as lung fibrosis and skin scarring *via* the nonclassical Wnt/Ca$^{2+}$ pathway (*Singla et al., 2023*; *Trinh-Minh et al., 2024*). Notably, the downregulation of *WNT5A* expression levels may hinder fibrosis progression through diminishing local immune responses following trauma (*Xue et al., 2022*), indicating potential as a diagnostic and prognostic marker of IUA. *BIRC5* functions as an antiapoptotic protein that enhances cell survival through inhibiting apoptosis as well as promotes tissue repair by sustaining fibroblasts during the fibrotic process (*Horowitz et al., 2012*). The downregulation of *BIRC5* expression levels leads to reduced fibrosis and compromised tissue repair. *GDF7* is a member of the *TGF*-β family that is primarily involved in neural and muscular differentiation. Although *GDF7* has received less attention in fibrotic diseases, *GDF7* may play a crucial role in tissue fibrosis through influencing fibroblast differentiation (*Kong et al., 2023*). Furthermore, *GDF7* downregulation may adversely affect tissue repair and fibrosis progression.

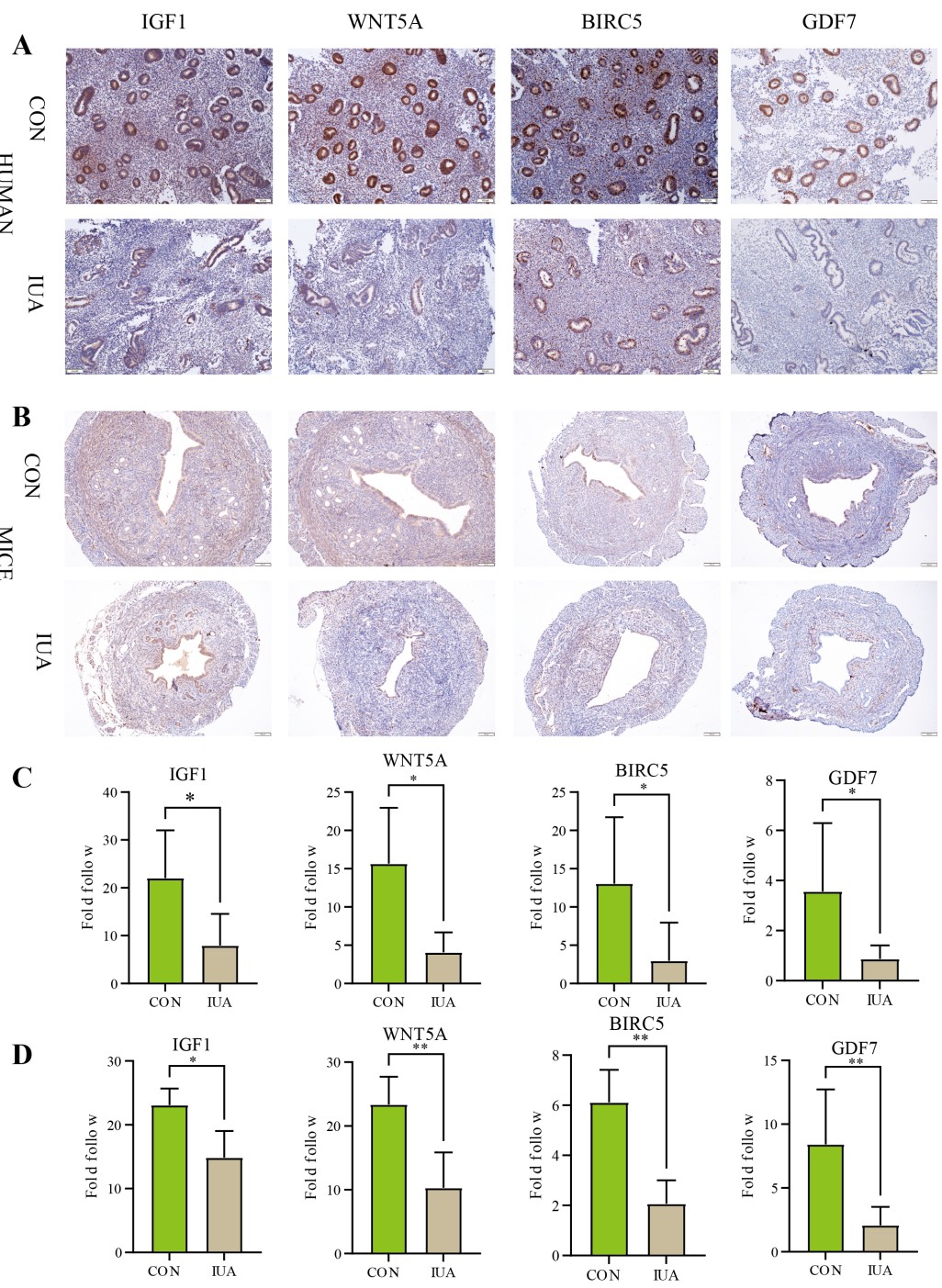

**Figure 6** **Immunohistochemical validation of hub protein expression in intrauterine adhesion.** (A) Representative IHC images showing protein expression levels of IGF1, WNT5A, BIRC5, and GDF7 in human endometrial tissues from IUA patients *versus* healthy controls (scale bar: 100 μm). (B) Corresponding IHC results in a mouse IUA model. (C) Quantitative analysis of IHC staining intensity in human tissues. (D) Quantitative analysis of IHC staining intensity in mouse tissues. Statistical significance was determined by Mann–Whitney U test (*P < 0.05, **P < 0.01).

**Table 3** Statistical analysis of immunohistochemical staining intensity for IGF1, WNT5A, BIRC5, and GDF7 in human and murine endometrial tissues.

| Genes | Species | Sample size (n$_1$ *vs* n$_2$) | *P*-value | Cliff's delta [95% CI] |
|---|---|---|---|---|
| IGF1 | Human | 5 *vs* 8 | 0.0109 | −0.85 [−28.829, −3.274] |
| | Mice | 5 *vs* 5 | 0.0317 | −0.77 [−6.877, −0.305] |
| WNT5A | Human | 5 *vs* 5 | 0.0159 | −0.92 [−20.789, −2.919] |
| | Mice | 5 *vs* 5 | 0.0079 | −1 [−20.407, −6.677] |
| BIRC5 | Human | 5 *vs* 8 | 0.0295 | −0.75 [−20.680, −2.802] |
| | Mice | 5 *vs* 5 | 0.0079 | −1 [−5.844, −2.554] |
| GDF7 | Human | 5 *vs* 7 | 0.0303 | −0.75 [−20.680, −2.802] |
| | Mice | 5 *vs* 5 | 0.0079 | −1 [−12.035, −0.884] |

**Notes.**

n$_1$, Normal control(CON) group; n$_2$, Intrauterine adhesion (IUA) group.

In conclusion, these genes play roles in fibrosis, cellular repair, and immune regulation. These genes may be suitable as biomarkers and therapeutic targets for IUA and thus warrant further investigation.

## Role of bioinformatics methods in exploring pathogenesis of IUA

Bioinformatics methods can be used to integrate and analyze multiomics data, providing tools for comprehensively understanding the molecular mechanisms of IUA. These approaches may aid in elucidating the molecular mechanisms underpinning IUA, facilitating the discovery of biomarkers and the development of personalized therapies (*Liang et al., 2024*). We identified the genes closely associated with the pathological processes of IUA using GSEA. The results of GO and KEGG pathway enrichment analyses identified the biological functions of these genes and the signaling pathways involved. We used WGCNA to identify the key hub genes in IUA through constructing gene coexpression networks, through which we determined their potential roles in pathological processes and established regulatory networks. Integrating these methodologies identified potentially suitable molecular targets for diagnosing and treating IUA. These methods are powerful tools that could be used in future mechanistic studies.

## Experimental validation

Although the results of bioinformatic analysis produced a large number of candidate genes, experimental validation was a step in ensuring the reliability of our findings. Our qPCR and IHC experiments validated the changes in the expression levels of some key genes in IUA, enhancing our confidence in the determined roles of these genes in IUA. However, the inconsistency between the experimental results and raw confidence analyses suggests that more samples and in-depth mechanistic studies are needed to explain these discrepancies. Further study will increase our understanding of the effects of the differences in the samples, species, and experimental conditions and technical errors on the results. Additional single- and multi-center studies may contribute to our understanding of the changes in the immune microenvironment during the pathogenesis of IUA.

The observed discrepancies between bioinformatic predictions and experimental validation, particularly for *TUBB3* and *PTGDS*, warrant careful consideration of species-specific biological and technical factors. *TUBB3* expression was upregulated and downregulated in the human and muring IUA endometrial specimens compared with the controls, respectively. However, the *PTGDS* expression level was up- and down-regulated in the human and murine IUA samples compared with the controls, respectively. These divergent trends may reflect (1) the fundamental differences in endometrial physiology between species, as murine models incompletely mimic human IUA pathophysiology; (2) technical limitations in cross-species primer design, exemplified by *CCL14* being undetectable in mice due to potential gene absence or primer incompatibility; and (3) microenvironmental variations, where human samples represent chronic IUA states *versus* acute murine injury responses. Such disparities underscore the need for species-matched validation when translating computational findings. The consistent validation of *IGF1*, *WNT5A*, *BIRC5*, and *GDF7* between the qPCR and IHC (human/mice $p < 0.05$) results reinforces their roles in IUA; however, the inconsistency in the genes highlights methodological challenges. The failure to amplify murine *CCL14* suggests gaps in murine genome annotation or expression sparsity, urging caution in assuming gene ortholog conservation. Researchers should (1) integrate multiomics methods (*e.g.*, single-cell RNA-seq) to clarify species-specific expression patterns, (2) optimize species-specific primer validation using long-read sequencing, and (3) incorporate patient-derived organoids to bridge translational gaps and address these issues. These strategies could mitigate the limitations posed by interspecies variability and technical artifacts, enabling further IUA biomarker discovery.

### Study limitations and future directions

This study has three principal limitations. (1) The restricted sample size of the GSE224093 dataset ($n = 14$) limited the statistical power for moderate-effect genes ($|\log2FC| < 1.5$), although this limitation was mitigated using FDR correction ($p < 0.05$) and external validation; (2) The modest validation cohort sizes (human: n = 5–8; mice: n = 5–8) provided adequate power (72–80%) for detecting large-effect biomarkers (Cohen's $d \geq 1.0$) but limited the sensitivity to moderate-effect targets ($d = 0.5$–0.8), as evidenced by the wider confidence intervals in the effect-size estimates. The core findings (*e.g.*, IGF1/WNT5A) demonstrated cross-species consistency; however, the small sample sizes precluded meaningful subgroup analyses and impact the generalizability of our findings to heterogeneous clinical populations. (3) Interspecies divergences—such as undetectable murine CCL14, opposing TUBB3/PTGDS regulation, and distinct inflammatory/fibrotic timelines—highlight the fundamental differences between surgical murine models and human IUA pathophysiology.

A tiered validation strategy is proposed to address these limitations. (1) Expanded multicenter cohorts (target n≥30/group) should be included, incorporating laser-capture microdissected endometrial compartments to enhance spatial resolution. (2) Humanized mouse models should be developed through the xenotransplantation of patient-derived endometrial stromal cells to bridge species gaps. (3) Single-cell multiomics (scRNA-seq +

spatial proteomics) should be used with paired human/murine samples to determine the cell-type-specific expression patterns confounding bulk analyses. The CCL14 discrepancy should be addressed through long-read sequencing to confirm murine gene absence *versus* technical detection failure. These approaches will help with systematically addressing the current limitations and identifying clinically actionable biomarkers and targeted therapies for IUA.

## CONCLUSIONS

We screened and validated the DEGs associated with IUA using network bioinformatics analysis. We determined the key roles of genes such as *IGF1*, *WNT5A*, *BIRC5*, and *GDF7* in the occurrence and development of IUA. These genes play important roles in fibrosis, immunomodulation, and tissue repair, indicating potential biomarkers for diagnosing and treating IUA. Future studies will be devoted to the clinical translation of these findings and developing new therapeutic strategies to improve the prognosis of patients with IUA.

**Abbreviations**

| | |
|---|---|
| **IUA** | intrauterine adhesions |
| **GEO** | Gene Expression Omnibus |
| **DEGs** | differentially expressed genes |
| **GO** | Gene Ontology |
| **KEGG** | Kyoto Encyclopedia of Genes and Genomes |
| **IRGs** | immune-related genes |
| **WGCNA** | weighted gene co-expression network analysis |
| **GSEA** | gene set enrichment analysis |
| **qPCR** | quantitative real-time PCR |
| **IHC** | immunohistochemistry |
| **EMT** | epithelial-mesenchymal transition |

## ACKNOWLEDGEMENTS

The authors thank all members of the Liu Dan team from Ningxia Medical University General Hospital for heir assistance in our work. All authors contributed equally to the study.

### Funding

This work was supported by the Natural Science Foundation of Ningxia (No. 2024AAC02068). The funders had no role in study design, data collection and analysis, decision to publish, or preparation of the manuscript.

### Grant Disclosures

The following grant information was disclosed by the authors:
The Natural Science Foundation of Ningxia: 2024AAC02068.

## Competing Interests

The authors declare there are no competing interests.

## Author Contributions

- Fengqing Lv conceived and designed the experiments, performed the experiments, analyzed the data, prepared figures and/or tables, authored or reviewed drafts of the article, and approved the final draft.
- Sang Luo conceived and designed the experiments, authored or reviewed drafts of the article, and approved the final draft.
- Fengjuan Xu performed the experiments, authored or reviewed drafts of the article, and approved the final draft.
- Yue Du performed the experiments, prepared figures and/or tables, and approved the final draft.
- Yiyun Bai performed the experiments, analyzed the data, authored or reviewed drafts of the article, and approved the final draft.
- Jingyi Zhang analyzed the data, prepared figures and/or tables, and approved the final draft.
- Xiaojie Zou analyzed the data, authored or reviewed drafts of the article, and approved the final draft.
- Dan Liu conceived and designed the experiments, prepared figures and/or tables, authored or reviewed drafts of the article, and approved the final draft.

## Human Ethics

The following information was supplied relating to ethical approvals (*i.e.*, approving body and any reference numbers):

The Ethics Review Committee of the General Hospital of Ningxia Medical University (ethics no.KYLL-2024-1527).

## Animal Ethics

The following information was supplied relating to ethical approvals (*i.e.*, approving body and any reference numbers):

The Ethical Review Committee of the General Hospital of Ningxia Medical University (approval no. IACUC-NYLAC-2023-136).

## Data Availability

The data is available at NCBI GEO: GSE224093.

## Clinical Trial Registration

The following information was supplied regarding Clinical Trial registration:

IACUC-NYLAC-2023-136.

## Supplemental Information

Supplemental information for this article can be found online at http://dx.doi.org/10.7717/peerj.20035#supplemental-information.

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
