# Peer review of "Immune-related hub genes in intrauterine adhesions: a bioinformatics approach"

_PeerJ, doi:10.7717/peerj.20035_

## Round 0.1 · original submission · Major Revisions

**Language Note:** The review process has identified that the English language must be improved. PeerJ can provide language editing services - please contact us at [email protected] for pricing (be sure to provide your manuscript number and title). Alternatively, you should make your own arrangements to improve the language quality and provide details in your response letter. – PeerJ Staff

Reviewer 1 ·

Basic reporting

The manuscript is written in ambiguous and unprofessional English with grammar mistakes, broken or run-on sentences, and misuse of punctuations throughout. For example, the first sentence under the Introduction section on Line 37-38 goes "IUA commonly known as Asherman syndrome, is a pathological condition..." where there lacks a comma between "IUA" and "commonly". Another example is that the entire paragraph on Line 112-118 was essentially written as a single run-on sentence with multiple independent clauses. Contrary to that, the sentence on Line 285-289 has no clause. The authors should seriously edit the current writings and greatly improve the readability of the article.

The authors also failed to discuss the figures in the main texts. Figure 3D was not mentioned at all any where in the article, while the entire Figure 4 was barely discussed (Line 281-289). Given such, it is unclear how the result data supports the hypotheses of the authors, hence article is not self-contained.

Experimental design

Admittedly, the authors have chosen an important topic with clinical relevance. However, the authors failed to motivate the research topic in the Introduction sections. On Line 38-46, the authors subsequently introduced the cause and formation of the disease, the clinical manifestations, and the current treatment options. Then, the authors used the term "however" that lead to the conclusion that the high recurrence rate of IUA still present challenges. Yet, it seems to the reviewer that there lacks a clear logical connection between the introduced backgrounds and the therapeutic limitations. In particular, it is unclear how understanding the immune microenvironment abnormalities can reduce disease recurrence rate and improve clinical successes.

The authors seem to have a fundamental misunderstanding of the Gene Set Enrichment Analysis (GSEA) method. On Line 69-70, the authors claimed that they obtained the differentially expressed genes by GSEA, while GSEA is simply not a differential gene calling tool. It appear to the reviewer that the authors are not equipped with the adequate amount knowledge of the bioinformatics tools they used in the article.

Validity of the findings

The authors need to improve the presentation of the statistical methods they use or refer to in the article. On one hand, whenever a p-value threshold is called upon, it is unclear whether the authors used un-adjusted or adjusted p-values. If un-adjusted, it is highly recommended that the authors should use adjusted p-values instead to avoid false positives. If adjusted, please explain which adjustment method was used, such as FDR, BH, etc. On the other hand, the authors also mentioned two different statistical testing methods on Line 248-250 depending on whether the underlying data is normally distributed or not. However, please explain how the normality (Gaussian distribution) of the data is determined.

Additional comments

No additional comments.

Reviewer 2 ·

Basic reporting

Strengths:
The manuscript is generally well-organised and follows the standard structure expected for a bioinformatics and translational study, with clear division into background, methodology, results, and discussion. Figures and tables are relevant and correspond appropriately to the narrative, with data visualisations that support key findings. The literature cited is appropriate and current, offering relevant context for the study’s rationale. The inclusion of both transcriptomic and experimental validation data adds depth to the analysis.

Major comments:
1. English language and grammar: The manuscript requires thorough revision for grammar, syntax, and clarity. Numerous typographical errors and ambiguous phrases limit readability. Example corrections include:
○ “its pathological mechanism remain unelucidated” → “the underlying pathological mechanisms remain unclear”
○ “biosignature analysis” to “bioinformatic analysis”
○ Consistently use “IUA” instead of variations such as “intrauterine adhesion” mid-sentence.
2. Abstract revisions needed:
○ The sentence “The GSE224093 dataset was extracted from the Gene Expression Omnibus (GEO) database” should be revised to describe the dataset directly, rather than citing the database. For example:
“A publicly available endometrial transcriptomic dataset comprising samples from patients with severe IUA and healthy controls was analysed.”
This avoids unnecessary reference to GEO in the abstract and improves scientific clarity.
3. Figures and legends: Some figure legends lack sufficient detail. For example:
○ Figure 3: Clarify terms like “soft threshold”, “module feature vector” for readers unfamiliar with WGCNA.
○ Heatmaps should clearly indicate control and IUA groups.
○ Statistical annotations (e.g. p-values) should be consistently shown and described in legends.
4. Terminology and formatting:
○ Several symbols (e.g. “g1” instead of “>1”, random “ÿ”, “#”) suggest OCR conversion artefacts or typesetting errors. These must be corrected throughout.
○ The term “Saddlebrow” appears in the methods without explanation. Please define or correct this.

Experimental design

1. Discrepancies between bioinformatic predictions and qPCR trends (e.g. TUBB3, PTGDS) are acknowledged but require deeper discussion.
2. The CCL14 gene could not be validated in mice. While the authors attribute this to species differences, additional justification should be provided—whether this is due to true absence or low expression.
3. Statements implying functional causality (e.g. “revealing key roles”) should be reworded to avoid overinterpretation in the absence of mechanistic validation.
4. The GSE224093 dataset contains only 14 samples (7 IUA, 7 control), limiting statistical power. This must be discussed explicitly.
5. GEO2R is used for DEG identification. This tool lacks the statistical robustness of pipelines such as DESeq2 or limma. Consider rerunning DEG analysis using a more rigorous statistical framework. Also, there is no specific description of the steps involved in the analysis, which model did you run? What filtering steps were applied to the data.
6. The rationale and methodology for gene filtering (DEGs ∩ IRGs ∩ WGCNA module genes) should be described more clearly and reproducibly.
7. Experimental validation sample sizes (especially IHC in mice and humans) are small. The manuscript should report whether power calculations were performed or justify the sample sizes more transparently.

Validity of the findings

The findings are supported by a an analysis framework, albeit a very basic GEO R one that needs to be improved. The identification of IGF1, WNT5A, BIRC5, and GDF7 as potential key regulators in the immune and fibrotic processes associated with IUA is biologically plausible and consistent with existing literature. The experimental validations in human and mouse samples, despite limited sample size, reinforce the interpretability of the computational results.

---

## Round 0.2 · accepted · Accept

Dear authors,

Thank you for addressing all the reviewers' concerns. Your work is now acceptable for publication.

Congratulations and thank you for submitting your work to PeerJ.

Best regards

Reviewer 1 ·

Basic reporting

The authors addressed each point the reviewer made in the previous round. No further comment is given.

Experimental design

no comment

Validity of the findings

no comment

Reviewer 2 ·

Basic reporting

All comments addressed

Experimental design

All comments addressed

Validity of the findings

All comments addressed

Additional comments

All comments addressed